# Optimization of Train Operation Planning with Full-Length and Short-Turn Routes of Virtual Coupling Trains

**Xu Zhou** [1,2,*] **, Fang Lu** [1] **and Liyu Wang** [1]

1   School of Traffic and Transportation, Beijing Jiaotong University, Beijing 100044, China
2   Traffic Control Technology Co., Ltd., Beijing 100070, China
*   Correspondence: 20125837@bjtu.edu.cn

**Abstract:** Virtual coupling uses wireless communication instead of mechanical coupling to ensure that trains are easily reconnected or disconnected. This technology can shorten the interval time between trains, give full play to the carrying capacity of lines, and improve the service level of urban rail transit. This paper optimizes the train operation plan with full-length and short-turn routes of virtual coupling trains by establishing a two-level optimization model. The upper model is used to minimize passenger travel time and enterprise operation cost, and the lower model to optimize the equilibrium of train load rate on short-turn routes. Meanwhile, a method based on the genetic algorithm is designed to solve the model. A case study of the Metro Line M has been carried out. The results can verify the efficiency and feasibility of the proposed method. The full-length and short-turn routes of virtual coupling trains can effectively reduce passenger travel time, enterprise operating cost and the number of vehicles, and improve the average load factor of the trains. Finally, sensitivity analyses are performed using three parameters which include departure frequency of the full-length train and short-turn train, starting and terminal station of short-turn route, and number of marshalled vehicles of the full-length train and short-turn train.

**Keywords:** urban rail transit; virtual coupling; full-length and short-turn routes; train operation plan; genetic algorithm

## 1. Introduction

The passenger flow is expanding in tandem with the extension of the urban rail transit network, resulting in a conflict between passenger travel demand and urban rail transit capacity supply. The traditional transportation organization mode has difficulty meeting current travel needs of passengers. Virtual coupling, however, can be utilized to provide technical support for flexible transportation organization mode. By better matching the needs of passengers with the transportation capacity, it can ensure the service frequency, improve the quality of transportation services, and increase the attraction to urban rail transit. In addition, it can also reduce the energy consumption of urban rail transit system.

## 2. Literature Review

Studies of virtual coupling technology have been widely carried out. Some scholars first studied the basic concepts and functions of virtual coupling. Konig et al. [1] and Ständer et al. [2] discussed the basic concept of virtual coupling, and verified the effectiveness and security of its functions. Goikoetxea et al. [3] introduced the technology required to achieve virtual coupling in Shift2Rail and explained that it was beneficial cost reduction. Gómez et al. [4] focused on the decentralization of the communication between moving trains and investigated the viability under realistic conditions. Virtual coupling is valuable in improving the carrying capacity of the line. It can also shorten train operation intervals and improve operation control efficiency. Schumann [5] simulated railway operations with virtual coupling in the Shinkansen scenario, which showed the possibility to increase

line capacity. Liu [6] proposed the multi-agent system (MAS) to control the operation of a virtually coupled train group. Felez et al. [7] showed that the virtual coupling concept substantially reduces headway and distance between trains while guaranteeing safe separation between two consecutive trains at any instant. Liu et al. [8] put forward a new method that considers the distribution of passenger flow to improve transportation efficiency. Flammini [9] and Aoun [10] investigated demand trends and operational scenarios of virtual coupling, Aoun [11] also analyzed its advantages and limitations through two extensive surveys. Bai et al. [12] analyzed the typical operation process using virtual coupling technology and provided a formula for calculating the station interval time.

Virtual coupling has new requirements for train control systems. Chen [13] summarized the redundant rule between the train-to-train direct communication and the existing communication. The results show that the automatic adjustment strategy inside the train formation can make the train operation efficient and keep the train tracking in a steady state. Song et al. [14] proposed a method of speed limit curve calculation based on relative braking distance, which may provide some reference for the next generation train control systems. Cui et al. [15] analyzed the key technical requirements of virtual coupling application in the train control system, and designed the architecture of a train control system using virtual coupling technology. Li [16] and Yu [17] researched train control methods of virtual coupling oriented to dynamic coupling.

The optimization problem of train operation plans for full-length and short-turn routes is usually defined as a mathematical optimization problem, for which models and algorithms have been designed. Wang [18], Wei [19], and Deng et al. [20] set up the optimization model for minimizing the travel time of passengers and the operating cost of enterprises, respectively. Liu [21] focused on the passenger flow of urban rail lines during peak hours and established an optimization model for the train operation scheme of full-length and short-turn routes. Xu et al. [22] and Duan [23] established an optimization operation scheme model based on the analysis of passengers' choice behavior of different routes. Considering fairness factors, Yao et al. [24] established an optimization model of full-length and short-turn routes with the goal of minimizing the travel delay of all passengers. Chen et al. [25] proposed a collaborative full-length and short-turning plan and joint multi-station control of passenger flow. Blanco et al. [26] propose a model for line planning and timetabling from a cost-oriented and a passenger-oriented perspective. Ren et al. [27] established a combined two-step model of train formation optimization and real-time station control. The load factor of the trains is also an important goal to be considered in the design of operation schemes. Dai et al. [28] and Zhang [29] analyzed the dynamic demand of passenger flow and established an optimization model to increase the average load factor of trains. Xu [30] and Wu [31] proposed an optimization method of a train operation scheme of full-length and short-turn routes considering the balance of load rate. Liu et al. [32] established an optimization model aiming at minimizing passenger waiting time, fixed operating cost of vehicles, and wasted capacity cost. Liao [33] established an optimization model, and comprehensively considered the constraints such as departure frequency, load factor of trains, and maximum waiting time that passengers could tolerate. Yang [34] constructed a train operation scheme model considering the passenger flow of the whole urban rail network, aiming at the optimal network efficiency. Rajabighamchi et al. [35] establish a model to optimize the multi-marshalling problem by minimizing the trains' vacant capacities. Zhao et al. [36] analyzed the passenger flow of an airport line and put forward three train operation plans based on virtual coupling trains. At present, there is almost no research on the train operation plan of virtual coupling trains. In addition, in the research on the train operation plan of full-length and short-turn routes, the number of marshalled vehicles is fixed. After applying virtual coupling trains, the number of marshalled vehicles can be changed during the operation. The full-length train and the short-turn train are quickly coupled or uncoupled, which makes full-length and short-turn route mode more flexible. This paper provides an idea for the optimization of the train operation plan of virtual coupling trains. Other researchers can refer to the ideas

of this paper and conduct research on transportation organization modes such as express and local trains, so as to contribute to improving the transportation efficiency and service level of urban rail transit.

In this study, a two-level optimization model is proposed. The upper model is established to minimize passenger travel time and enterprise operation cost, and the lower model to optimize the equilibrium of train load rate on short-turn routes. Meanwhile, a method based on the genetic algorithm is designed to solve the model. Different from previous studies, this paper mainly focuses on the train operation scheme based on virtual coupling. Finally, a case study of the Metro Line M is carried out. The results verify the efficiency and feasibility of the proposed method.

The rest of this paper is organized as follows. Section 2 describes the proposed problem and assumptions. In Section 3, the two-level optimization model is established. A solution method based on a genetic algorithm is designed to search for the optimal solution in Section 4. Section 5 uses a case to demonstrate the effectiveness and superiority of the proposed model and algorithm. Finally, the conclusions and future work are presented in the last section.

## 3. Description of the Problem and Assumptions

### 3.1. Analysis Problem

The traditional full-length and short-turn route mode needs to determine the number of marshalled vehicles of full-length trains according to the passenger flow of the whole line, but with the application of virtual coupling trains, the transportation organization mode of full-length and short-turn routes can be more flexible. The number of marshalled vehicles of short-turn trains can be determined according to the passenger flow in the non-overlapping sections of full-length and short-turn routes. Therefore, virtual coupling trains can use a smaller number of marshalled vehicles of full-length trains, which reduces the cost to enterprises. In this paper, the full-length train and the short-turn train are coupled at the turn-back station of the short-turn routing, and the two trains operate on the short-turn routing to provide more transport capacity. When the two trains arrive at another turn-back station of the short-turn routing, they will be uncoupled. The full-length train continues to run forward, while the short-turn train turns back at the turn-back station, forming the full-length and short-turn routings. The transportation organization mode of virtual coupling trains can reduce enterprise operating costs and improve service levels.

This paper aims at optimizing the train operation plan with full-length and short-turn routes of virtual coupling trains. It establishes a two-level optimization model. The upper model is used to minimize passenger travel time and enterprise operation cost, and the lower model to optimize the equilibrium of train load rate on short-turn routes. The upper model adopts the departure frequency of the train and the location of the turn-back station of the short-turn as the decision variables, and the lower model takes the number of marshalled trains as the decision variable. Since the objective function of the upper model and that of the lower model are not in the same order of magnitude, they are considered separately. The two-layer optimization model is designed according to the two optimization objectives, which can better take into account the factors of cost and full load rate.

As shown in Figure 1, there are $N$ stations in the line, and the direction of trains running from station 1 to station $N$ is upward. The short-turn train only runs between station $a$ and station $b$. In this paper, stations 1 to $a$ are regarded as section I. Then, stations $a$ to $b$ are recorded as section II, and stations $b$ to N are recorded as section III. $f_1$ is departure frequency of the full-length train, and $f_2$ is departure frequency of the short-turn train. $f_1$, $f_2$, $a$, $b$ are decision variables of the upper model. $n_1$ is the number of marshalled vehicles of the full-length train, and $n_2$ is the number of marshalled vehicles of the short-turn train. $n_1$ and $n_2$ are decision variables of the lower model.

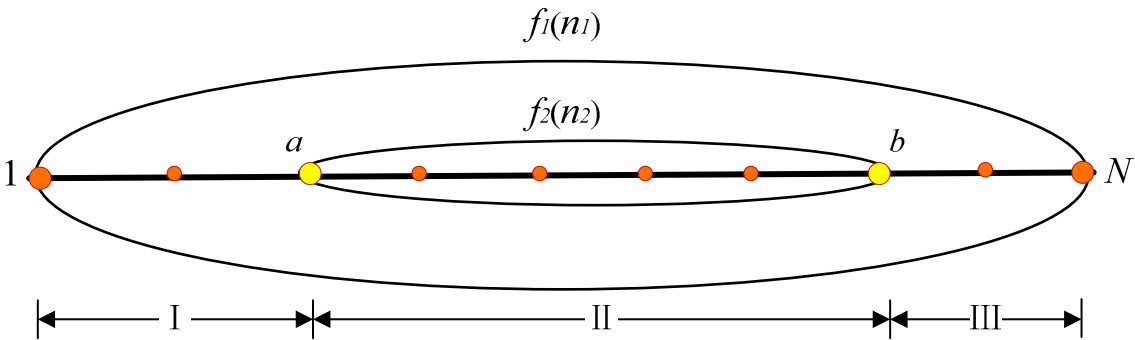

**Figure 1.** Schematic Diagram of Train Plan with Full-length and Short-turn Routes.

*3.2. Assumptions*

Train operation plans for full-length and short-turn routes that can accommodate the specific passenger flow pattern on this route are determined by the inter-station distances, safe headway, the passenger origin–destination (OD) flow distribution, and other factors.

The following basic assumptions are made:

(1) Passengers entering each station during the period examined in this study follow a uniform distribution. This paper does not consider the situation of passenger detention.
(2) All trains depart from the originating station at equal time intervals.
(3) When passengers can take trains of full-length routing and trains of short-turn routing, the proportion of people taking different trains is determined according to the operation proportion of different trains.
(4) All trains stop at each station, and the operation time in the up direction is equal to that in the down direction.
(5) All stations have the ability to turn back, and the time for turning back is equal.

**4. Model Formulation**

Based on the above modeling assumptions, a two-level optimization model for full-length and short-turn routes of virtual coupling trains is established. The upper model minimizes passenger travel time and enterprise operation cost, with the departure frequency of the train and the location of the turn-back station of the short-turn route as the decision variables. The lower model optimizes the equilibrium of load factor between full-length trains and short-turn trains, when the trains are running on short-turn routes. The lower model takes the number of marshalled trains as the decision variable.

*4.1. Model Parameters*

Tables 1–3 summarize the main model parameters and their definitions.

**Table 1.** Definition of decision variables.

| Parameter | Definition |
|:---:|:---:|
| $f_1$ | Departure frequency of the full-length train |
| $f_2$ | Departure frequency of the short-turn train |
| $a$ | Starting station of short-turn route |
| $b$ | Terminal station of short-turn route |
| $n_1$ | Number of marshalled vehicles of the full-length train |
| $n_2$ | Number of marshalled vehicles of the short-turn train |

**Table 2.** Definition of upper-layer model parameters.

| Parameter | Definition |
|---|---|
| $q_{ij}$ | The passenger flow from station $i$ to station $j$ |
| $T_k$ | Calculation period (1 h) |
| $L_{1N}$ | Total length of the line |
| $L_{ab}$ | Length of short-turn route |
| $m, n$ | A random positive integer |
| $f_0$ | Minimum departure frequency |
| $f_m$ | Maximum departure frequency |
| $Z$ | Number of vehicles used |
| $Z_0$ | Number of vehicles used in a single train operation plan |
| $R_i$ | Operation time of train in section $i$ |
| $S_j$ | Stop time of the train at station $j$ |
| $t_z$ | Turn-back time of trains |
| $\lceil\ \rceil$ | Round up |

**Table 3.** Definition of the lower model parameters.

| Parameter | Definition |
|---|---|
| $q_{ij}$ | The passenger flow from station $i$ to station $j$ |
| $\beta_1$ | Proportion with the second category of passengers taking the full-length train |
| $\beta_2$ | Proportion with the second category of passengers taking the short-turn train |
| $\beta'_1$ | Proportion with the third category of passengers taking the full-length train |
| $\beta'_2$ | Proportion with the third category of passengers taking the short-turn train |
| $p$ | Probability of passengers choosing not to get on the short-turn train |
| $n_0$ | Maximum number of marshalled vehicles |
| $\gamma'_{\max}$ | Maximum load factor of the trains |
| $\gamma_{\min}$ | Lower limit with load factor of the trains |
| $\gamma_{\max}$ | Upper limit with load factor of the trains |
| $C$ | Capacity of each train |

*4.2. Upper Model Construction*

4.2.1. Objective Functions

To minimize passenger travel time and enterprise operation cost, the upper model adopts the departure frequency of the train and the location of the turn-back station of the short-turn route as the decision variables. Both $T_w$ and $L_s$ are minimized.

1.      Calculation of travel cost of passengers $T_w$

As passengers are insensitive to time when they are on the train, this paper takes the waiting time as the travel cost of passengers. Due to the difference in the section of the departure station and the terminal station of the passenger travel, the types of trains that passengers can take are also different. Therefore, passengers can be classified according to the type of train they take, and then the waiting time for each type of passenger is calculated. This paper mainly classifies passengers in the up direction, as shown in Figure 2, which can be divided into 6 kinds.

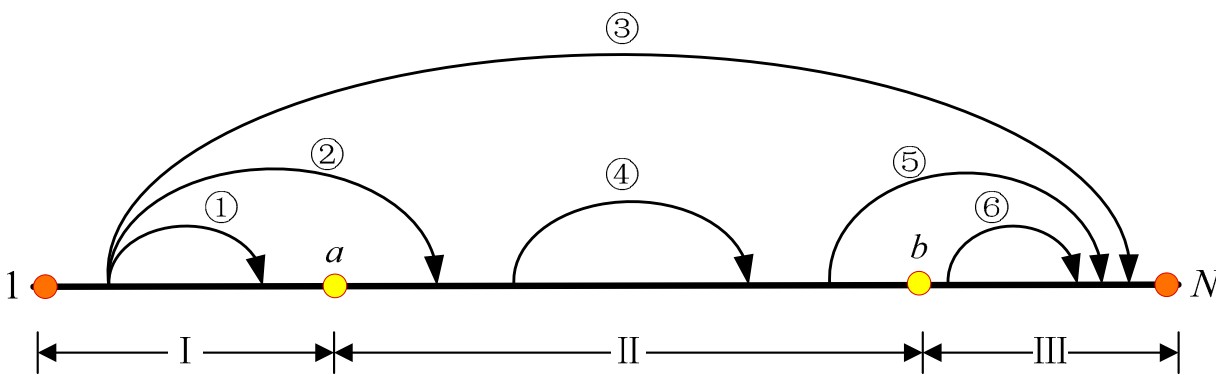

**Figure 2.** Passenger classification diagram.

(1) The first kind of passenger

The starting station and terminal station of the first kind of passenger are in section I. This kind of passenger can only take the full-length train.

(2) The second kind of passenger

The starting station of the second kind of passenger is in section I and their terminal station is in section II. This kind of passenger can only take the full-length train.

(3) The third kind of passenger

The starting station of the third kind of passenger is in section I and their terminal station is in section III. They can only take the full-length train.

(4) The fourth kind of passenger

The starting station and terminal station of the fourth kind of passenger are in section II. This kind of passenger can take the full-length train or short-turn train.

(5) The fifth kind of passenger

The starting station of the fifth kind of passenger is in section II and terminal station of this kind of passenger is in section III. They can take the full-length train to the terminal station directly, or take the short-turn train first and then change to take the full-length train.

(6) The sixth kind of passenger

The starting station and terminal station of the sixth kind of passenger are in section III. They can only take the full-length train.

In the upper model, it is considered that the fifth kind of passenger will only take the full-length train. Based on this analysis, this paper regards passengers who can only take the full-length train as the first category of passengers, including the first, second, third, fifth and sixth kinds of passenger. Passengers who can take the full-length train or short-turn train are regarded as the second category of passenger, including the fourth kind of passenger.

Passenger flow of the first category of passenger can be calculated using Equations (1)–(3):

$$Q_1 = Q_1^+ + Q_1^-, \tag{1}$$

$$Q_1^+ = \sum_{i=1}^{a-1} \sum_{j=i+1}^{N} q_{ij} + \sum_{i=a}^{b-1} \sum_{j=b+1}^{N} q_{ij} + \sum_{i=b}^{N-1} \sum_{j=i+1}^{N} q_{ij}, \tag{2}$$

$$Q_1^- = \sum_{i=2}^{a} \sum_{j=1}^{i-1} q_{ij} + \sum_{i=a+1}^{b} \sum_{j=1}^{a-1} q_{ij} + \sum_{i=b+1}^{N} \sum_{j=1}^{i-1} q_{ij}. \tag{3}$$

$Q_1^+$ and $Q_1^-$ are the passenger flow of the first category of passenger in the up direction and down direction.

Passenger flow of the second category of passenger can be calculated using Equations (4)–(6):

$$Q_2 = Q_2^+ + Q_2^-, \tag{4}$$

$$Q_2^+ = \sum_{i=a}^{b-1} \sum_{j=i+1}^{b} q_{ij}, \tag{5}$$

$$Q_2^- = \sum_{i=a+1}^{b} \sum_{j=a}^{i-1} q_{ij}. \tag{6}$$

$Q_2^+$ and $Q_2^-$ are the passenger flow of the second category of passenger in the up direction and down direction.

Departure interval of trains on the full-length route can be obtained by the following formula:

$$I_1 = T_k / f_1. \tag{7}$$

Departure interval of trains on the short-turn route can be obtained by the following formula:

$$I_2 = T_k / (f_1 + f_2). \tag{8}$$

Taking half of the departure interval as the average passenger waiting time, the waiting time of the first kind of passenger can be obtained by the following formula:

$$t_{w1} = Q_1 \cdot I_1 / 2. \tag{9}$$

The waiting time of the second kind of passenger can be calculated as follows:

$$t_{w2} = Q_2 \cdot I_2 / 2. \tag{10}$$

The total passenger waiting time is as follows:

$$T_w = t_{w1} + t_{w2}. \tag{11}$$

The objective function of passenger travel cost is as follows:

$$\min W_1 = T_w. \tag{12}$$

2. Calculate the cost of enterprise operation $L_s$

In this paper, the running kilometer of trains is regarded as the operating cost of the enterprise. It can be calculated using Equations (13)–(15):

$$L_s = L_1 + L_2, \tag{13}$$

$$L_1 = 2 \cdot L_{1N} \cdot f_1 \cdot n_1 + 2 \cdot L_{ab} \cdot f_1 \cdot n_2, \tag{14}$$

$$L_2 = 2 \cdot L_{ab} \cdot f_2 \cdot n_2. \tag{15}$$

The objective function of passenger travel cost is designed as follows:

$$\min W_2 = L_s. \tag{16}$$

### 4.2.2. Constraints

1. Constraints on the departure frequency

In order to facilitate the drawing of the train diagram, it should be ensured that the departure frequency of the full-length train is an integer multiple of the departure frequency of the short-turn train, or vice versa.

$$f_1 = m \cdot f_2 \tag{17}$$

$$f_2 = n \cdot f_1 \tag{18}$$

2. Constraints on the starting station and terminal station of short-turn route

The starting station and terminal station of the short-turn route are selected from all stations of the line.

$$1 \le a \le b \le N \tag{19}$$

3. Constraints on the minimum departure frequency

In order to ensure the service level of urban rail transit, the maximum departure interval should not be too long. In this paper, the maximum departure interval is 6 min, so the minimum departure frequency is 10 pairs/h.

$$f_1 \ge f_0 \tag{20}$$

4. Constraints on the maximum departure frequency

The maximum frequency of departure should not exceed the maximum capacity of the line.

$$f_1 + f_2 \le f_m \tag{21}$$

5. Constraints on the number of vehicles used

The number of vehicles used in the train operation plan with full-length and short-turn routes of virtual coupling trains shall not be more than that in the single train operation plan:

$$Z \le Z_0. \tag{22}$$

The number of vehicles used in the train operation plan with full-length and short-turn routes of virtual coupling trains $Z$ can be calculated as follows:

$$Z = (n_1 + n_2) \cdot \lceil T_1 \cdot f_1 \rceil + n_2 \cdot \lceil T_2 \cdot f_2 \rceil. \tag{23}$$

Turnaround time of the full-length train can be calculated as follows:

$$T_1 = 2 \cdot \left( \sum_{i=1}^{N-1} R_i + \sum_{j=2}^{N} S_j + t_z \right). \tag{24}$$

Turnaround time of the short-turn train can be calculated as follows:

$$T_2 = 2 \cdot \left( \sum_{i=a}^{b-1} R_i + \sum_{j=a+1}^{b} S_j + t_z \right). \tag{25}$$

*4.3. Lower Model Construction*

4.3.1. Objective Functions

The lower model aims to optimize the equilibrium of load factor between full-length short-turn train for the trains running on short-turn routes. The lower model takes the number of marshalled trains as the decision variable.

Since the objective function of this model considers the load factor equilibrium of trains running on short-turn routes, it mainly considers passengers who will pass through short routes, including second, third, fourth, and fifth kinds of passenger. This paper regards passengers who can only take the full-length train as the first category of passenger, including the second and third kinds of passenger. Passengers who can take the full-length train or short-turn train are regarded as the second category of passenger, including the fourth kind of passenger. The fifth kind of passenger, who can either take the full-length train to the terminal station directly, or take the short-turn train first and then change to take the full-length train, is regarded as the third category of passenger.

Passenger flow of the first category of passenger can be calculated using Equations (26)–(28):

$$M_1 = M_1^+ + M_1^-,\tag{26}$$

$$M_1^+ = \sum_{i=1}^{a-1}\sum_{j=a}^{N} q_{ij},\tag{27}$$

$$M_1^- = \sum_{i=b+1}^{N}\sum_{j=1}^{b} q_{ij}.\tag{28}$$

$M_1^+$ and $M_1^-$ are the passenger flow of the first category of passenger in the up direction and down direction.

Passenger flow of the second category of passenger can be calculated using Equations (29)–(31):

$$M_2 = M_2^+ + M_2^-,\tag{29}$$

$$M_2^+ = \sum_{i=a}^{b-1}\sum_{j=i+1}^{b} q_{ij},\tag{30}$$

$$M_2^- = \sum_{i=a+1}^{b}\sum_{j=a}^{i-1} q_{ij}.\tag{31}$$

$M_2^+$ and $M_2^-$ are the passenger flow of the second category of passenger in the up direction and down direction.

Passenger flow of the third second category of passenger can be calculated using Equations (32)–(34):

$$M_3 = M_3^+ + M_3^-,\tag{32}$$

$$M_3^+ = \sum_{i=a}^{b-1}\sum_{j=b+1}^{N} q_{ij},\tag{33}$$

$$M_3^- = \sum_{i=a+1}^{b}\sum_{j=1}^{a-1} q_{ij}.\tag{34}$$

$M_3^+$ and $M_3^-$ are the passenger flow of the third category of passenger in the up direction and down direction.

For the second category of passenger, the probability of taking a full-length train or short-turn train is equal to the proportion of the train departure frequency of full-length trains and short-turn trains.

The probability of the second category of passenger taking the full-length train can be obtained by the following formula:

$$\beta_1 = f_1/(f_1 + f_2).\tag{35}$$

Probability of the second category of passenger taking the short-turn train can be obtained by the following formula:

$$\beta_2 = 1 - \beta_1.\tag{36}$$

Passengers in the third category who choose to take a short-turn train first have to transfer halfway to a full-length train. As a result, some passengers prefer not to board the train and instead continue to wait for the full-length train when short-turn trains arrive. In this paper, it is assumed that the probability of passengers choosing not to get on the short-turn train is $p$, which is taken as 0.2.

Probability of the third category of passenger taking the full-length train can be calculated as follows:

$$\beta'_1 = \beta_1 + \beta_2 \cdot p. \tag{37}$$

Probability of the third category of passenger taking the short-turn train can be calculated as follows:

$$\beta'_2 = \beta_2 \cdot (1 - p). \tag{38}$$

The equilibrium of load factor between the full-length train and short-turn train can be calculated as follows:

$$\varphi = \left( \frac{M_1 + \beta_1 \cdot M_2 + \beta'_1 \cdot M_3}{2 \cdot f_1 \cdot (n_1 + n_2) \cdot C} - \frac{\beta_2 \cdot M_2 + \beta'_2 \cdot M_3}{2 \cdot f_2 \cdot n_2 \cdot C} \right)^2. \tag{39}$$

The objective function of the lower function is designed as follows:

$$\min W_3 = \varphi. \tag{40}$$

### 4.3.2. Constraints

1. Constraint on the number of marshalled vehicles

The maximum number of marshalled vehicles is mainly limited by the length of the platform. Since this paper assumes that the minimum number of marshalled vehicles with virtual coupling trains is 2, the minimum sum of marshalled vehicles with full-length trains and short-turns train is 4.

$$4 \le n_1 + n_2 \le n_0 \tag{41}$$

2. Constraint of the maximum load factor of the trains

$$\gamma_{\min} \le \gamma'_{\max} \le \gamma_{\max} \tag{42}$$

Maximum load factor of the trains can be calculated as follows:

$$\gamma'_{\max} = \max\{\gamma_1^{x+}, \gamma_1^{x-}, \gamma_2^{x+}, \gamma_2^{x-}\}. \tag{43}$$

The load factor of trains in the up direction of section X of the full-length routes can be calculated as follows:

$$\gamma_1^{x+} = \begin{cases} \sum\limits_{i=1}^{x} \sum\limits_{j=x+1}^{N} q_{ij} / (f_1 \cdot n_1 \cdot C), 1 \le x < a \\ \dfrac{\sum\limits_{i=1}^{a-1} \sum\limits_{j=x+1}^{N} q_{ij} + \beta_1 \cdot \sum\limits_{i=a}^{x} \sum\limits_{j=x+1}^{b} q_{ij} + \beta'_1 \cdot \sum\limits_{i=a}^{x} \sum\limits_{j=x+1}^{N} q_{ij}}{f_1 \cdot (n_1 + n_2) \cdot C}, a \le x < b. \\ \sum\limits_{i=1}^{x} \sum\limits_{j=x+1}^{N} q_{ij} / (f_1 \cdot n_1 \cdot C), b \le x < N \end{cases} \tag{44}$$

The load factor of trains in the down direction of section X of the full-length routes can be calculated as follows:

$$\gamma_1^{x-} = \begin{cases} \sum\limits_{i=x+1}^{N} \sum\limits_{j=1}^{x} q_{ij} / (f_1 \cdot n_1 \cdot C), 1 \le x < a \\ \dfrac{\sum\limits_{i=b+1}^{N} \sum\limits_{j=1}^{x} q_{ij} + \beta_1 \cdot \sum\limits_{i=x+1}^{b} \sum\limits_{j=a}^{x} q_{ij} + \beta'_1 \cdot \sum\limits_{i=x+1}^{N} \sum\limits_{j=1}^{a-1} q_{ij}}{f_1 \cdot (n_1 + n_2) \cdot C}, a \le x < b. \\ \sum\limits_{i=x+1}^{N} \sum\limits_{j=1}^{x} q_{ij} / (f_1 \cdot n_1 \cdot C), b \le x < N \end{cases} \tag{45}$$

The load factor of trains in the up direction of section X of the short-turn routes can be calculated as follows:

$$\gamma_2^{x+} = \left[ \beta_2 \cdot \sum_{i=a}^{x} \sum_{j=a}^{x} q_{ij} + \beta_2' \cdot \sum_{i=a}^{x} \sum_{j=b+1}^{N} q_{ij} \right] / (f_2 \cdot n_2 \cdot C). \tag{46}$$

The load factor of trains in the down direction of section X of the short-turn routes can be calculated as follows:

$$\gamma_2^{x-} = \left[ \beta_2 \cdot \sum_{i=x+1}^{b} \sum_{j=a}^{x} q_{ij} + \beta_2' \cdot \sum_{i=x+1}^{N} \sum_{j=1}^{a-1} q_{ij} \right] / (f_2 \cdot n_2 \cdot C). \tag{47}$$

## 5. Solution Algorithms

The normalization method is used to transform the two-objective optimization problem described by the upper model to a single-objective optimization problem, which can be solved using a genetic algorithm. In addition, there are few feasible solutions of the lower model, so the lower model can be solved by enumeration.

### 5.1. The Procedure of the Algorithm

The two-layer optimization model is designed according to the two optimization objectives, which can better take into account the factors of cost and full load rate. As a bi-level model, the lower level's decision variables are the parameters for the upper-level model. $f1, f2, a, b$ are decision variables of the upper model, and they are also the variables that make up the objective function of the lower model. Firstly, the genetic algorithm is used to obtain the optimal solution of the upper model. Then, the optimal frequencies and locations of the turn-back station from the upper model are substituted into the lower model to solve its optimal solution. The procedure of the algorithm is provided below.

Step1: The upper model is converted into a single-objective optimization problem by normalization. The weights of each objective function are determined based on the objective function values of the single-routing operation plan.

$$\min W = \alpha_1 \cdot W_1 + \alpha_2 \cdot W_2 \tag{48}$$

$T_{W0}$ is the waiting time for passengers of the single train operation scheme, $L_{S0}$ is the train running distance under the single routing mode. $\alpha_1$ and $\alpha_2$ are the weights of the two objective functions, which can be calculated as follows:

$$\alpha_1 \cdot T_{W0} = \alpha_2 \cdot L_{S0}, \tag{49}$$

$$\alpha_1 + \alpha_2 = 1. \tag{50}$$

Step2: The upper model can be solved using a genetic algorithm. First, we need to determine the parameters required by the upper model, including crossover probability, mutation probability, and initial number of marshalled vehicles. The initial population has 100 individuals. The evolution is 50 generations. The initial number of marshalled vehicles is $n1$ and $n2$, both of which are 4. The key steps are shown below.

(1) Individual coding: One-dimensional binary coding is adopted. The upper model established in this paper has 4 decision variables, so each chromosome consists of 4 gene segments. Among them, the number of genes corresponding to the departure frequency of the full-length train and short-turn train is 4, the number of genes corresponding to the starting station and terminal station of the short-turn route is 5. In summary, the total number of genes per chromosome is 18. As shown in Figure 3, $f1$ is 12, $f2$ is 3, $a$ is 9, and $b$ is 13.

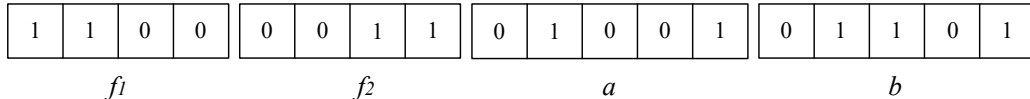

**Figure 3.** Chromosome coding.

(2) Selection strategy: The tournament algorithm is used to select individuals. The reciprocal of the objective function is taken as the fitness function. Penalty function is used to screen infeasible solutions. Before calculating the objective function, decode the chromosome first, and then check whether this chromosome meets the constraint. The objective function corresponding to the chromosome that does not meet the constraint is a larger value. As shown in Figure 4, $f1$ is 16, $f2$ is 6, $a$ is 13, and $b$ is 9. This chromosome does not meet the restrictions of Formula (17) and Formula (19), so this chromosome has a larger objective function value.

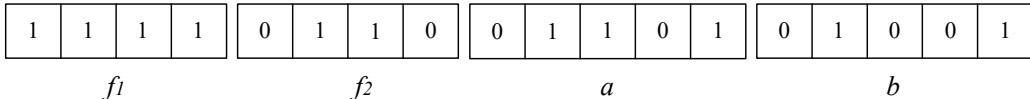

**Figure 4.** Chromosome coding.

(3) Single-point crossover: Two adjacent chromosomes are grouped together for crossover. If the chromosomes are odd, the last chromosome does not cross. Randomly select a gene bit and generate a random number. If the random number is less than the crossover probability, crossover is performed to generate a new chromosome. Single-point crossover is shown in Figure 5.

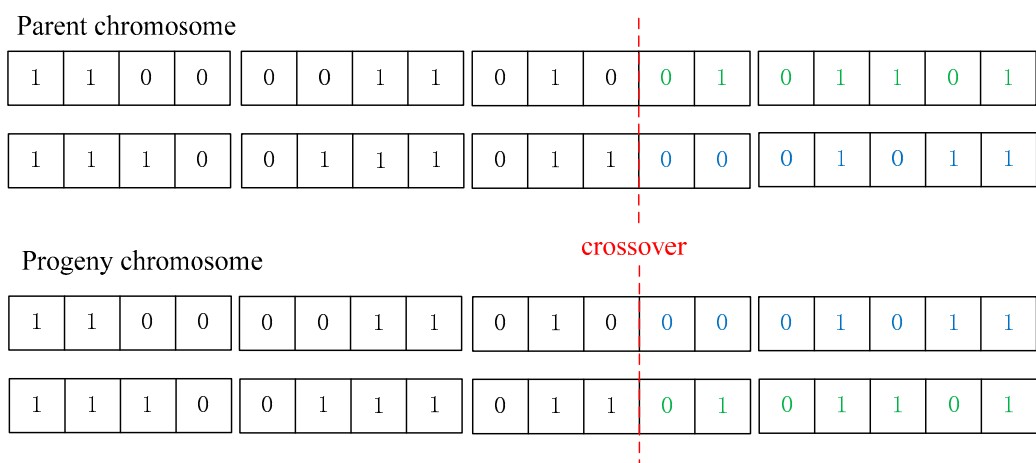

**Figure 5.** Single-point crossover.

(4) Basic bit mutation: Randomly select a gene bit and generate a random number, if the random number is less than the mutation probability, the gene bit will change. As shown in Figure 6, the mutation principle is that 0 becomes 1, and 1 becomes 0.

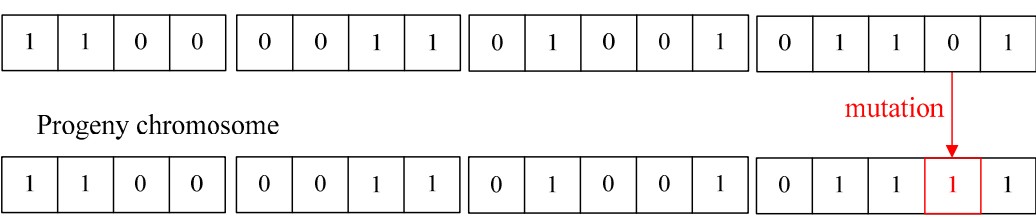

**Figure 6.** Basic bit mutation.

(5) Termination criterion: Repeat (2) to (4) 50 times, compare the optimal chromosomes in 50 generations, and take the chromosome with the maximum value of fitness function as the output result.

Step3: The optimal solution of the upper model is input into the lower model as a known parameter.

Step4: Enumeration is applied to the optimal solution of the lower model. Save the optimal solution of the upper and lower models.

Step5: After repeating step 2 to step 4 30 times, the algorithm ends.

The flowchart of the algorithm is shown in Figure 7.

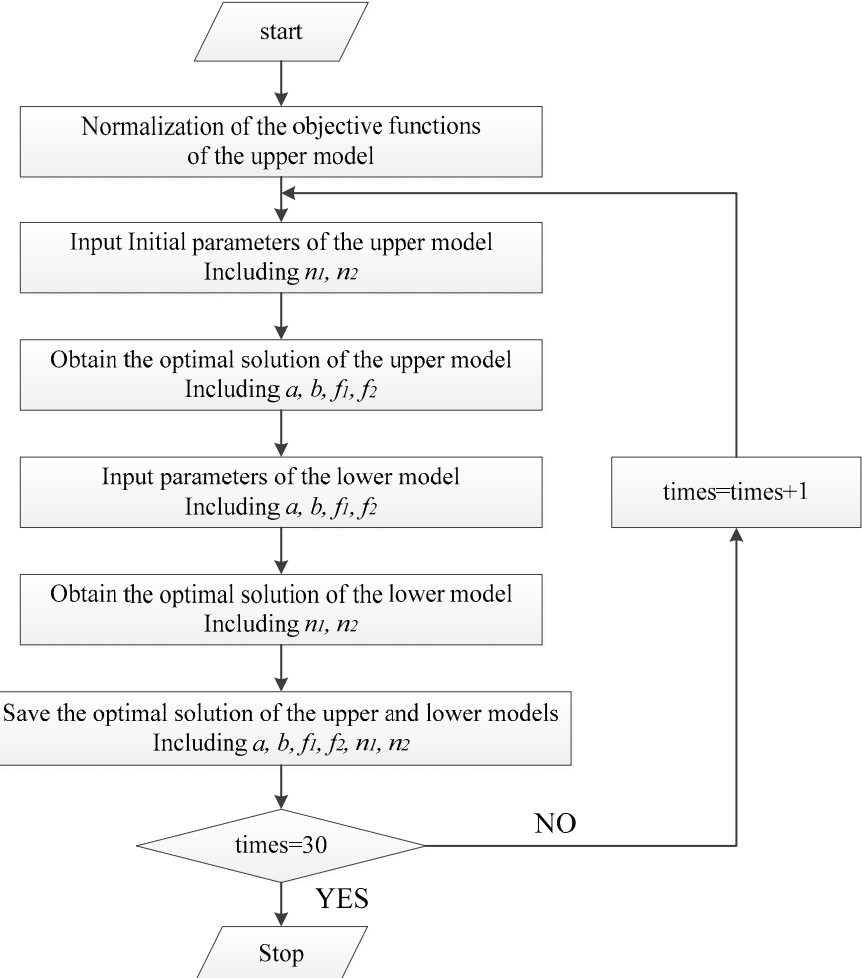

**Figure 7.** The flowchart of the algorithm.

### 5.2. Determine Parameters

Virtual passenger flow data is used to test the algorithm. The objective function of the upper model includes the waiting time of passengers and the running kilometers of trains. NSGA2 can be used to solve the non-dominated optimal solution of two objectives. The mutation probability is 0.02. As shown in Figure 8, the crossover probabilities are 0.5, 0.6, 0.7, and 0.8, respectively. We can choose the appropriate solution according to the demand, but there is no fixed standard to explain which solution is better. Therefore, the normalization method is used to transform the two-objective optimization problem to a single-objective optimization problem.

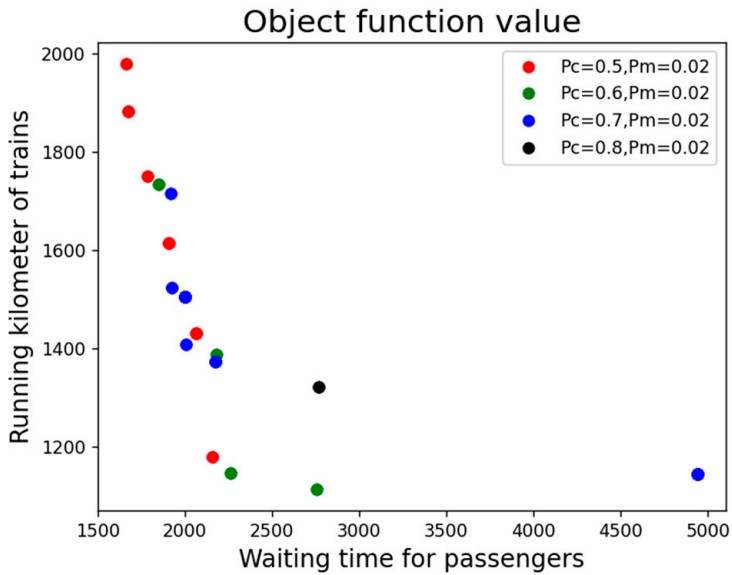

**Figure 8.** Object function value of NSGA2.

After being transformed into a single-objective optimization problem, the GA is used to solve it. The mutation probability is 0.02. The crossover probabilities are 0.5, 0.6, 0.7, and 0.8, respectively. The fitness function value is shown in Figure 9. When the crossover probability is 0.7, the variation range of the objective function value is relatively large, and finally it can reach a better value. Therefore, the mutation probability of this paper is 0.7, and the mutation probability is taken as 0.02. Evolution algebra is 50.

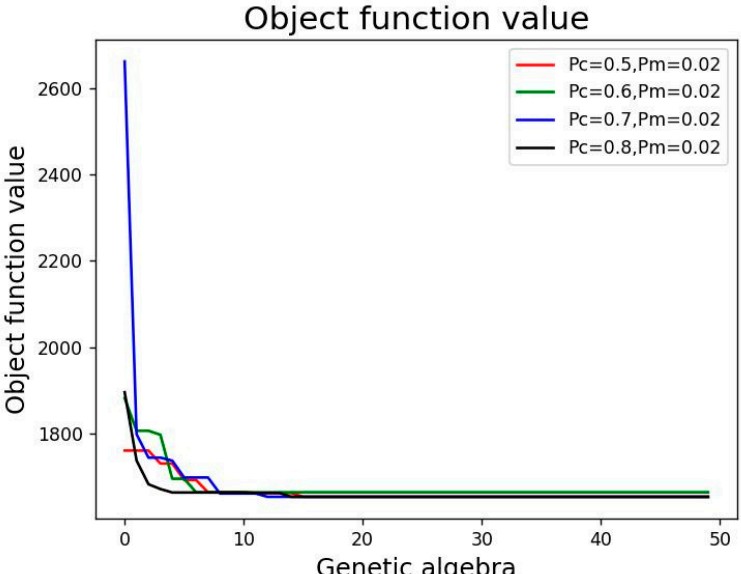

**Figure 9.** Object function value of GA.

## 6. Case Study on Metro M

The main purpose of this case is to study the advantages of full-length and short-turn routes of virtual coupling trains. It is assumed that each station has the ability to turn back, so a virtual case of Metro Line M is adopted. The model proposed in this paper can be applied to any line. A case study of Metro Line M during the morning peak period was carried out to find an optimal train operation plan with full-length and short-turn routes of virtual coupling trains. After obtaining the train operation plan, we select five evaluation indicators of waiting time for passengers, running kilometer of trains, number of vehicles used, maximum load factor of trains, and average load factor of trains to evaluate each

plan. The line stretches 29.24 km across 21 stations, with an average inter-station distance of 1.46 km. The overview of Metro Line M is shown in Table 4.

**Table 4.** Overview of Metro Line M.

| Inter-Station Segment No. | Inter-Station Distance/m | Inter-Station Running Time/s | Inter-Station Segment No. | Inter-Station Distance/m | Inter-Station Running Time/s |
|---|---|---|---|---|---|
| 1 | 1.15 | 89 | 11 | 1.48 | 113 |
| 2 | 1.38 | 106 | 12 | 1.80 | 138 |
| 3 | 1.76 | 135 | 13 | 1.63 | 125 |
| 4 | 1.52 | 116 | 14 | 1.26 | 96 |
| 5 | 1.30 | 100 | 15 | 1.76 | 135 |
| 6 | 1.32 | 101 | 16 | 1.13 | 87 |
| 7 | 1.43 | 110 | 17 | 1.29 | 96 |
| 8 | 1.63 | 125 | 18 | 1.39 | 107 |
| 9 | 1.24 | 95 | 19 | 2.11 | 150 |
| 10 | 1.27 | 97 | 20 | 1.42 | 109 |

### 6.1. Parameter Values

Table 5 summarizes the values of various model parameters.

**Table 5.** The Value of Parameters.

| Symbol | Definition | Value | Unit |
|---|---|---|---|
| $f_0$ | Minimum departure frequency | 10 | pairs/h |
| $f_m$ | Maximum departure frequency | 36 | pairs/h |
| $Z_0$ | Number of vehicles used in a single train operation plan | 180 | vehicles |
| $S_j$ | Stop time of the train at station $j$ | 40 | s |
| $t_Z$ | Turn-back time of trains | 100 | s |
| $n_0$ | Maximum number of marshalled vehicles | 10 | vehicles |
| $\gamma_{\min}$ | Lower limit with load factor of the trains | 60 | % |
| $\gamma_{\max}$ | Upper limit with load factor of the trains | 120 | % |
| $C$ | Capacity of each train | 240 | persons |
| $\alpha_1$ | Weight of total travel time | 0.686 | - |
| $\alpha_2$ | Weight of total operating trains | 0.314 | - |

### 6.2. Cross-Section Passenger Flow Data

Table 6 summarizes the passenger flow volume in each inter-station segment during morning peak hours (7:30–8:30). The maximum passenger flow volume of inter-station segments in the up direction is 24,407 persons/h, and the maximum passenger flow volume of inter-station segments in the down direction is 21,048 persons/h. This line applies 6B trains with 17 pairs/h departure frequency in the morning peak hours, and the rated capacity of each train is 240 passengers.

**Table 6.** Passenger Flow Volume in Each Inter-station Segment of Metro Line M.

| Inter-Station Segment No. | Sectional Passenger Volume in the Up Direction (Person/h) | Sectional Passenger Volume in the Down Direction (Person/h) |
|---|---|---|
| 1 | 1637 | 1013 |
| 2 | 2938 | 2049 |

**Table 6.** *Cont.*

| Inter-Station Segment No. | Sectional Passenger Volume in the Up Direction (Person/h) | Sectional Passenger Volume in the Down Direction (Person/h) |
|---|---|---|
| 3 | 4115 | 3232 |
| 4 | 4760 | 5010 |
| 5 | 13,575 | 7199 |
| 6 | 18,221 | 9699 |
| 7 | 22,388 | 12,664 |
| 8 | 24,407 | 15,301 |
| 9 | 23,003 | 16,192 |
| 10 | 22,105 | 18,223 |
| 11 | 21,759 | 19,891 |
| 12 | 19,556 | 20,846 |
| 13 | 17,714 | 21,048 |
| 14 | 16,483 | 20,586 |
| 15 | 15,837 | 17,674 |
| 16 | 13,852 | 16,603 |
| 17 | 11,653 | 15,745 |
| 18 | 8513 | 13,192 |
| 19 | 5313 | 2815 |
| 20 | 2585 | 1412 |

*6.3. Calculation Results and Analysis*

6.3.1. Optimal Solution in Two-Layer Model

An optimal train operation plan for Metro Line M during morning peak hours can be obtained by solving the model with a genetic algorithm. The solution process of the algorithm is to use the genetic algorithm to solve the optimal solution of the upper model (the optimal solution in 50 generations), and then use the enumeration method to solve the optimal solution of the lower model. This is a complete solution process. We repeat this process 30 times to save the 30 optimal solutions. Selected solutions that meet the objectives in the resulting analysis are listed in the table. The best four from the 30 optimal outcomes are selected: the minimum passenger waiting time, the minimum running kilometers of trains, the optimal objective function of the upper model, and the optimal objective function of the lower model. Table 7 summarizes four results. Table 8 summarizes a traditional train plan of full-length and short-turn routes and a single train plan. Then, from five evaluation indexes, it analyzes the train plan of full-length and short-turn routes of virtual coupling trains, traditional train plan of full-length and short-turn routes, and single train plan.

**Table 7.** Train Plan of Full-length and Short-turn Routes of Virtual Coupling Trains.

| | | Minimum Passenger Waiting Time | Minimum Running Kilometers of Trains | Optimal Upper Model | Optimal Lower Model |
|---|---|---|---|---|---|
| Decision variables | Departure frequency of the full-length train (pairs/h) | 12 | 10 | 12 | 15 |
| | Departure frequency of the short-turn train (pairs/h) | 12 | 10 | 12 | 5 |
| | Starting station of short-turn route (stations) | 4 | 5 | 5 | 5 |
| | Terminal station of short-turn route (stations) | 19 | 19 | 19 | 18 |
| | Number of marshalled vehicles of the full-length train (vehicles) | 2 | 2 | 2 | 2 |
| | Number of marshalled vehicles of the short-turn trains (vehicles) | 4 | 4 | 4 | 4 |

**Table 7.** *Cont.*

|  |  | Minimum Passenger Waiting Time | Minimum Running Kilometers of Trains | Optimal Upper Model | Optimal Lower Model |
|---|---|---|---|---|---|
| Objective functions | Objective function of upper model | 2857.99 | 2904.73 | 2811.09 | 2868.66 |
|  | Objective function of lower model | 0.16 | 0.18 | 0.12 | 0.03 |
| Evaluation index | Waiting time for passengers (h) | 2178.46 | 2681.88 | 2234.90 | 2508.91 |
|  | Running kilometers of trains (km) | 4342.56 | 3391.60 | 4069.92 | 3654.60 |
|  | Number of vehicles used (vehicles) | 180 | 144 | 176 | 168 |
|  | Maximum load factor of train (%) | 97.0 | 114.9 | 95.8 | 120.0 |
|  | Average load factor of train (%) | 60.5 | 76.7 | 63.9 | 75.7 |

**Table 8.** Traditional Train Plan of Full-length and Short-turn Routes and Single Train Plan.

|  |  | Minimum Passenger Waiting Time | Minimum Running Kilometers of Trains | Optimal Upper Model | Single Train Operation Plan |
|---|---|---|---|---|---|
| Decision variables | Departure frequency of the full-length train (pairs/h) | 15 | 11 | 14 | 17 |
|  | Departure frequency of the short-turn train (pairs/h) | 5 | 11 | 7 | - |
|  | Starting station of short-turn route (stations) | 8 | 6 | 8 | - |
|  | Terminal station of short-turn route (stations) | 15 | 15 | 15 | - |
|  | Number of marshalled vehicles of the full-length train (vehicles) | 6 | 6 | 6 | 6 |
|  | Number of marshalled vehicles of the short-turn train (vehicles) | 6 | 6 | 6 | - |
| Objective functions | Objective function of upper model | 3683.39 | 3793.49 | 3712.38 | 3744.74 |
|  | Objective function of lower model | 0.67 | 0.51 | 0.77 | - |
| Evaluation index | Waiting time for passengers (h) | 2610.66 | 2896.18 | 2740.58 | 2728.12 |
|  | Running kilometers of trains (km) | 6027.00 | 5753.88 | 5835.48 | 5965.78 |
|  | Number of vehicles used (vehicles) | 162 | 162 | 168 | 180 |
|  | Maximum load factor of train (%) | 114.9% | 112.4% | 115.2% | 99.7% |
|  | Average load factor of train (%) | 72.7% | 65.8% | 68.5% | 55.2% |

6.3.2. Comparison of the Results

Passenger Waiting Time

All train operation plans with virtual coupling trains can reduce passenger waiting time, as shown in Figure 10. As shown in Table 7, the first train operation plan of virtual coupling trains has the minimum passenger waiting time. As shown in Table 8, the first traditional train plan of full-length and short-turn routes has the minimum passenger waiting time. The first train operation plan of virtual coupling trains can reduce the waiting time of passengers by 16.6% and 20.2%, respectively, compared with the first traditional train plan of full-length and short-turn routes and the single train plan. It shows that the train operation plan of virtual coupling trains can greatly reduce the waiting time of passengers and improve the service level.

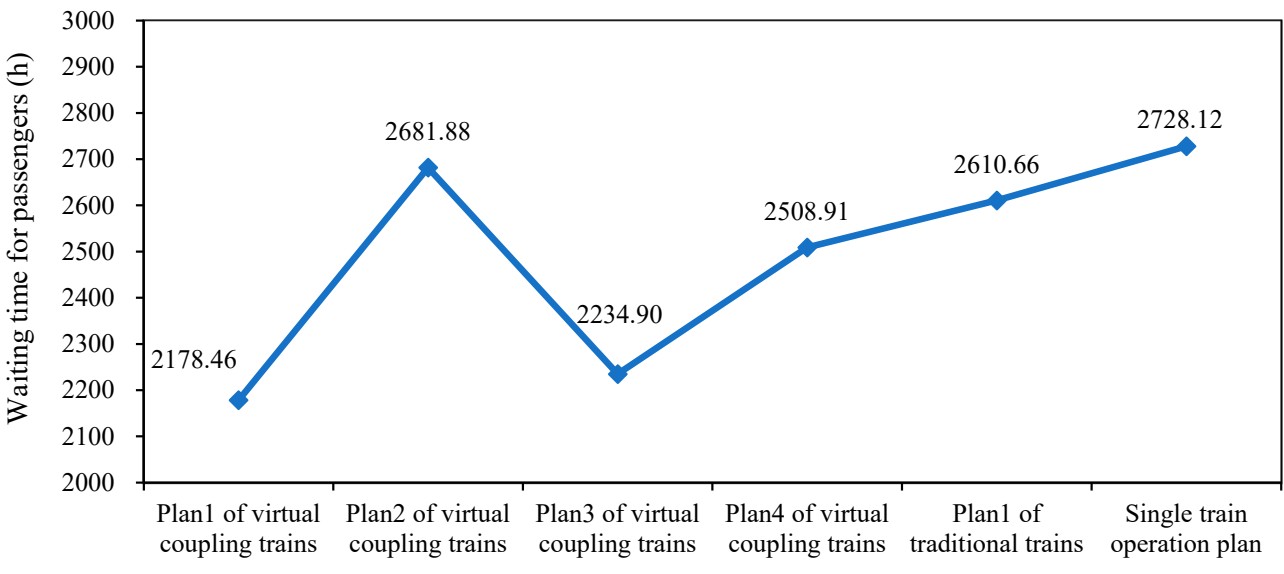

**Figure 10.** Comparison of Passenger waiting time.

Kilometers of Train Operation

Each train operation plan with virtual coupling trains can reduce kilometers of train operation, as shown in Figure 11. As shown in Table 7, the second train operation plan with virtual coupling trains has the minimum running kilometers of trains. In the second train operation plan with virtual coupling trains, there are two marshalling trains on the full-length routes and four marshalling trains on the short-turn routes. This scheme mainly increases the departure frequency of short-turn routes, but decreases the departure frequency of full-length routes. Therefore, this scheme has reduced the smallest number of running kilometers of trains by 43.1% and the smallest number of vehicles is reduced by 20% when compared to the single train operation scheme. As shown in Table 8, the second traditional train plan of full-length and short-turn routes has the minimum running kilometers of trains. Compared with the single train operation scheme, the running kilometers of trains are also reduced, but it is still worse than the second train operation plan with virtual coupling trains. The second train operation plan with virtual coupling trains can reduce running kilometers of trains by 41.1% compared with the second traditional train plan of full-length and short-turn routes. It shows that the train operation plan of virtual coupling trains can greatly reduce the running kilometers of trains and enterprise cost.

Maximum and Average Load Factor of Train

As shown in Figure 12, the train operation plan with virtual coupling trains can improve the maximum and average load factor of trains. As shown in Table 7, the full load rate index of the second train operation plan of virtual coupling trains is the best. The maximum full load rate of this scheme does not exceed the limit, and the average full load rate is high. As shown in Table 8, the first traditional train plan of full-length and short-turn routes has the best maximum and average load factor of trains, which can effectively improve the average load factor of trains. However, it is still worse than the second train operation plan with virtual coupling trains. The second train operation plan of virtual coupling trains can improve the average load factor of trains by 1.4% and 20.4%, respectively, compared with the first traditional train plan of full-length and short-turn routes and the single train plan. It shows that the train operation plan of virtual coupling can improve the utilization rate of train transportation capacity.

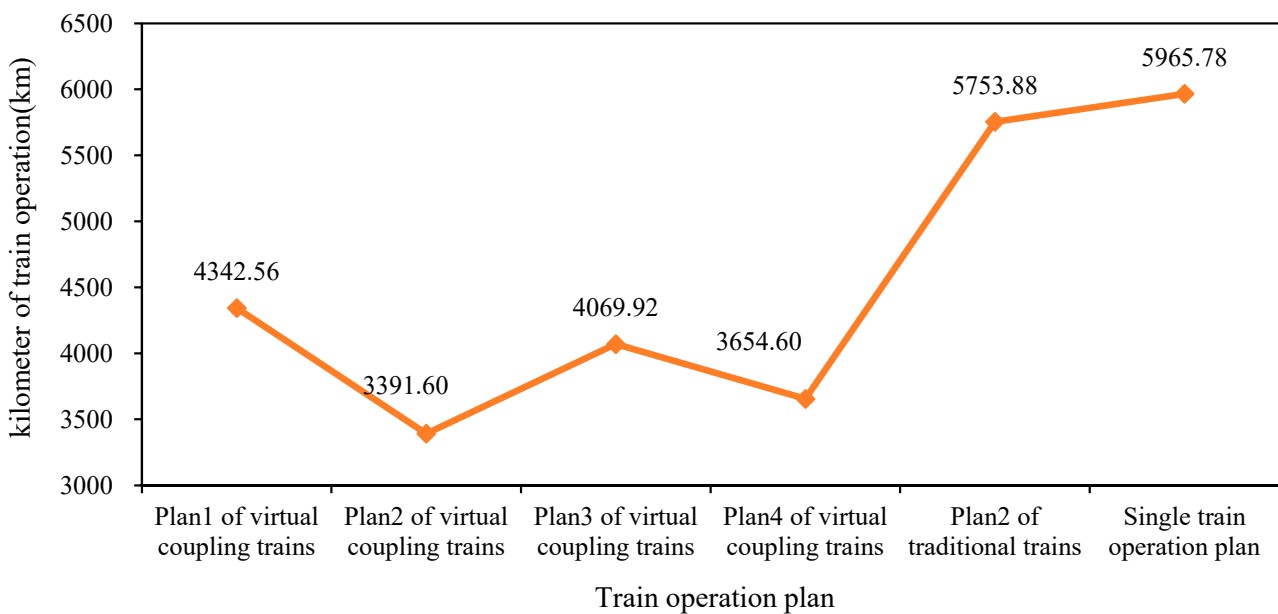

**Figure 11.** Comparison of Kilometers of Train Operation.

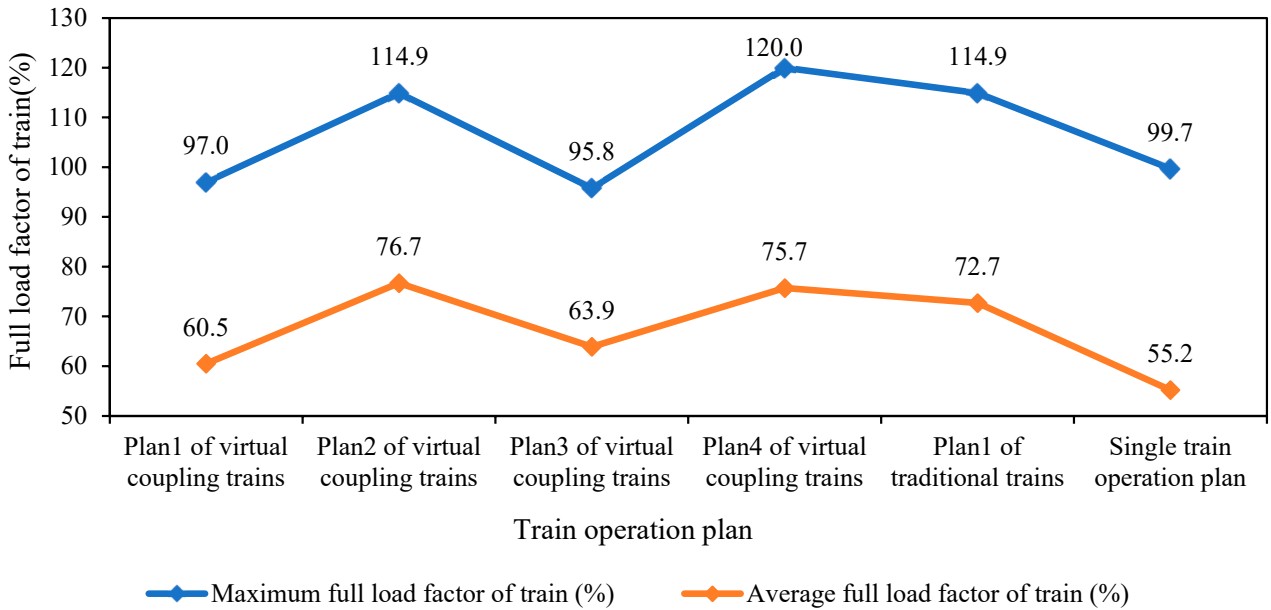

**Figure 12.** Comparison of Load Factor of Train.

### 6.4. Sensitivity Analyses

Sensitivity analyses were performed on the developed model to identify the critical parameters and their effects on the objective function. Sensitivity analyses are performed using three parameters which include departure frequency of the full-length train and short-turn train, starting and terminal station of the short-turn route, and number of marshalled vehicles of the full-length train and short-turn train. All of these parameters have a high impact on the behavior of the value of the objective function. Regarding each mentioned parameter, four plans are considered.

For the departure frequency of the full-length train and short-turn train, the sensitivity analyses are performed, and the results are provided in Table 9. To further illustrate values of the upper and lower objective function, a double coordinate axis is set in Figure 8. In Figure 13, the primary axis is the value of the upper objective function, and the secondary axis is the value of the lower objective function. Based on the results, as the departure

frequencies of the full-length train and short-turn train increase, the values of upper and lower objective functions are decreased. As shown in Table 9, the increase in departure frequency can greatly reduce the waiting time of passengers and the load factor of trains, so the value of the objective function can be reduced. However, using a larger departure frequency will increase the number of vehicles used. The staff need to adopt the appropriate departure frequency according to the passenger flow and the number of available vehicles.

**Table 9.** Sensitivity analyses of departure frequency.

| | | Plan 1 of Virtual Coupling Trains | Plan 2 of Virtual Coupling Trains | Plan 3 of Virtual Coupling Trains | Plan 4 of Virtual Coupling Trains |
|---|---|---|---|---|---|
| Decision variables | Departure frequency of the full-length train (pairs/h) | 9 | 10 | 11 | 12 |
| | Departure frequency of the short-turn train (pairs/h) | 9 | 10 | 11 | 12 |
| | Starting station of short-turn route (stations) | 5 | 5 | 5 | 5 |
| | Terminal station of short-turn route (stations) | 19 | 19 | 19 | 19 |
| | Number of marshalled vehicles of the full-length train (vehicles) | 2 | 2 | 2 | 2 |
| | Number of marshalled vehicles of the short-turn train (vehicles) | 4 | 4 | 4 | 4 |
| Objective functions | Objective function of upper model | 3002.65 | 2904.73 | 2843.97 | 2811.09 |
| | Objective function of lower model | 0.22 | 0.18 | 0.15 | 0.12 |
| Evaluation index | Waiting time for passengers (h) | 2979.86 | 2681.88 | 2438.07 | 2234.90 |
| | Running kilometers of trains (km) | 3052.44 | 3391.60 | 3730.76 | 4069.91 |
| | Number of vehicles used (vehicles) | 130 | 144 | 160 | 176 |
| | Maximum load factor of train (%) | 127.7% | 114.9% | 104.5% | 95.7% |
| | Average load factor of train (%) | 85.2% | 76.7% | 69.7% | 63.9% |

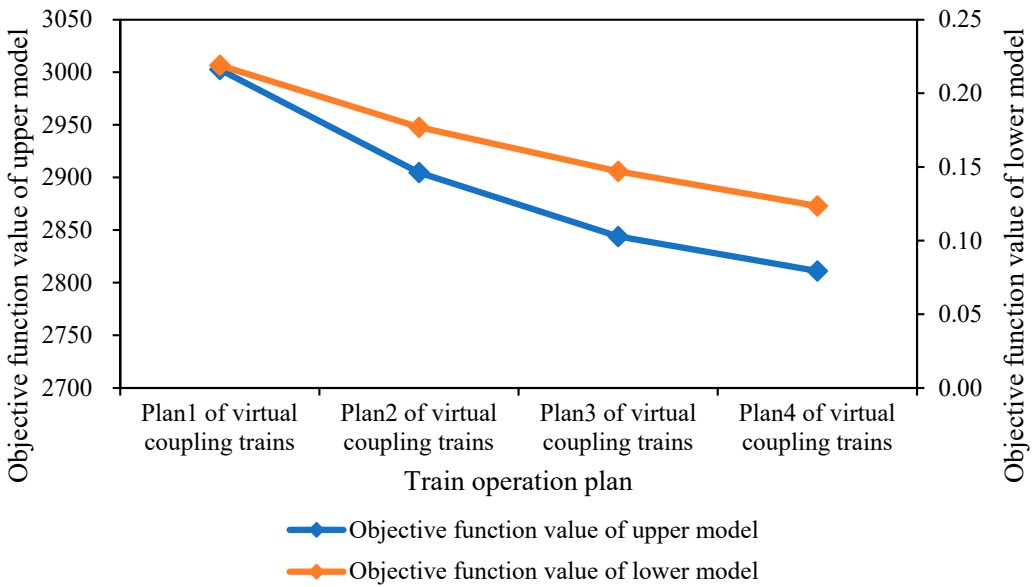

**Figure 13.** Sensitivity analyses of departure frequency.

To analyze the effect of the starting and terminal station of the short-turn route, four train operation plans are analyzed by varying this parameter's value, as seen in Table 10. Additionally, a double coordinate axis is set in Figure 8. With the increase in the length of the short-turn routing, the running kilometers of the train will be increased, and the waiting time of passengers will be greatly reduced. Therefore, on the whole, the value of the upper objective function will decrease. As shown in Figure 14, the upper objective function of plan 4 has the best value, but the lower objective function has the worst value. The staff need to consider the value of the upper and lower objective functions, the number of vehicles used, the full load rate of trains, and other indicators in combination with the passenger flow, and then select the appropriate starting and terminal station of the short-turn route.

**Table 10.** Sensitivity analyses of starting and terminal station of short-turn route.

| | | Plan 1 of Virtual Coupling Trains | Plan 2 of Virtual Coupling Trains | Plan 3 of Virtual Coupling Trains | Plan 4 of Virtual Coupling Trains |
|---|---|---|---|---|---|
| Decision variables | Departure frequency of the full-length train (pairs/h) | 10 | 10 | 10 | 10 |
| | Departure frequency of the short-turn train (pairs/h) | 10 | 10 | 10 | 10 |
| | Starting station of short-turn route (stations) | 5 | 4 | 5 | 4 |
| | Terminal station of short-turn route (stations) | 18 | 18 | 19 | 19 |
| | Number of marshalled vehicles of the full-length train (vehicles) | 2 | 2 | 2 | 2 |
| | Number of marshalled vehicles of the short-turn train (vehicles) | 4 | 4 | 4 | 4 |
| Objective functions | Objective function of upper model | 3062.05 | 3085.04 | 2904.73 | 2929.61 |
| | Objective function of lower model | 0.00 | 0.00 | 0.18 | 0.23 |
| Evaluation index | Waiting time for passengers (h) | 3005.68 | 2935.20 | 2681.88 | 2614.15 |
| | Running kilometers of trains (km) | 3185.20 | 3412.40 | 3391.60 | 3618.80 |
| | Number of vehicles used (vehicles) | 140 | 140 | 144 | 144 |
| | Maximum load factor of train (%) | 177.3% | 177.3% | 114.9% | 116.0% |
| | Average load factor of train (%) | 82.4% | 78.2% | 76.7% | 72.6% |

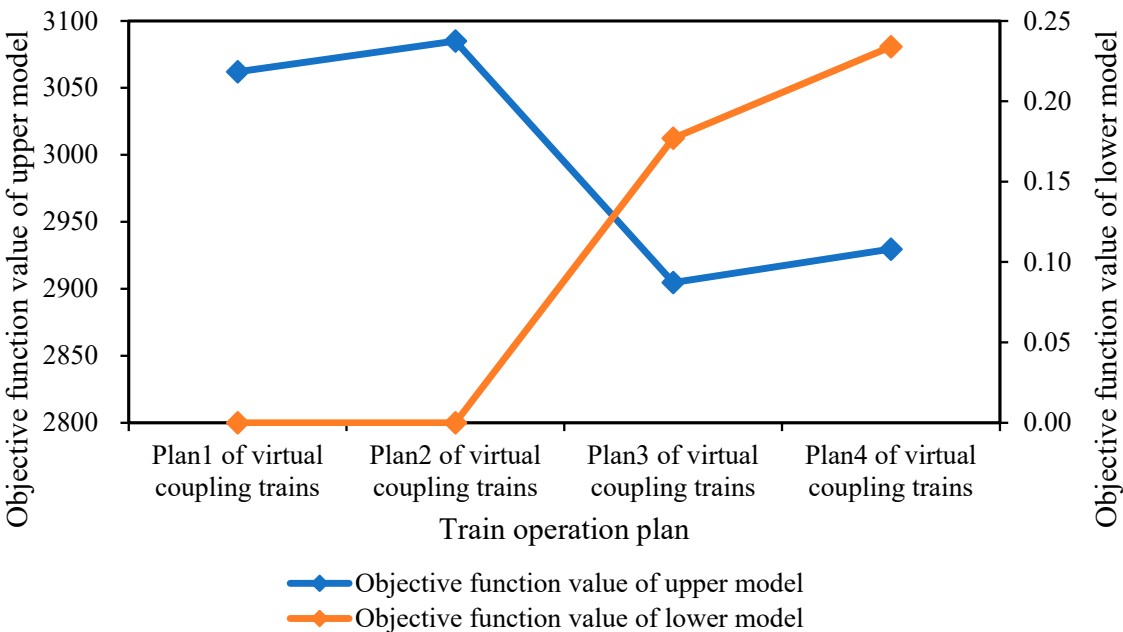

**Figure 14.** Sensitivity analyses of starting and terminal station of short-turn route.

The numbers of marshalled vehicles of the full-length train and short-turn train are also an important parameter. The sensitivity analysis results of the number of marshalled vehicles are shown in Table 11. It can be seen from Figure 15 that with the change in the number of marshalled vehicles, the values of the upper and lower objective functions change. Among them, the values of the upper and lower objective function of plan 1 are better, and the number of vehicles used is lower. The number of marshalled vehicles of the full-length train is two, and the number of marshalled vehicles of the short-turn train is four. The number of marshalled vehicles has a great influence on the value of the objective function and the evaluation index. Selecting the appropriate number of marshalled vehicles plays an important role in improving the service level of urban rail transit and reducing operating costs.

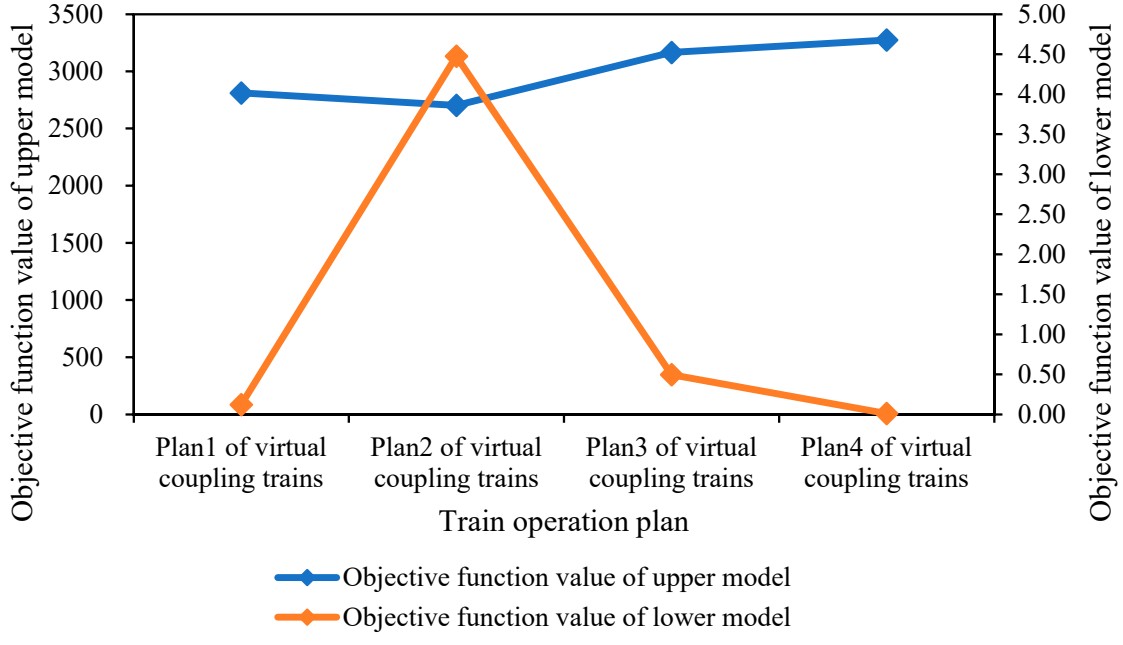

**Figure 15.** Sensitivity analyses of number of marshalled vehicles.

**Table 11.** Sensitivity analyses of number of marshalled vehicles.

| | | Plan 1 of Virtual Coupling Trains | Plan 2 of Virtual Coupling Trains | Plan 3 of Virtual Coupling Trains | Plan 4 of Virtual Coupling Trains |
|---|---|---|---|---|---|
| Decision variables | Departure frequency of the full-length train (pairs/h) | 12 | 12 | 12 | 12 |
| | Departure frequency of the short-turn train (pairs/h) | 12 | 12 | 12 | 12 |
| | Starting station of short-turn route (stations) | 5 | 5 | 5 | 5 |
| | Terminal station of short-turn route (stations) | 19 | 19 | 19 | 19 |
| | Number of marshalled vehicles of the full-length train (vehicles) | 2 | 4 | 4 | 2 |
| | Number of marshalled vehicles of the short-turn train (vehicles) | 4 | 2 | 4 | 6 |
| Objective functions | Objective function of upper model | 2811.09 | 2702.72 | 3164.83 | 3273.20 |
| | Objective function of lower model | 0.12 | 4.47 | 0.50 | 0.01 |
| Evaluation index | Waiting time for passengers (h) | 2234.90 | 2234.90 | 2234.90 | 2234.90 |
| | Running kilometers of trains (km) | 4069.92 | 3724.80 | 5196.48 | 5541.60 |
| | Number of vehicles used (vehicles) | 176 | 148 | 216 | 244 |
| | Maximum load factor of train (%) | 95.8% | 191.5% | 95.7% | 92.2% |
| | Average load factor of train (%) | 63.9% | 87.0% | 52.3% | 48.3% |

## 7. Conclusions

In this paper, a two-level optimization model for full-length and short-turn routes of virtual coupling trains is established. The upper-level model takes departure frequencies of full-length and short-turn trains and locations of short-turn mode turn-back stations as decision variables, and objective functions are to minimize both passenger travel time and enterprise operation cost. The lower-level model is an optimization model of the equilibrium of load factor, which is to determine the optimal formation plans. Meanwhile, a method based on the genetic algorithm is designed to solve the model. On the basis of previous research, this paper mainly focuses on train operation plan of full-length and short-turn routes with virtual coupling trains. A case of Metro Line M is used to show the reasonability and effectiveness of the proposed model. The model proposed in this paper can be applied to any line. The results show that passenger waiting time, kilometers of train operation, and number of vehicles used are significantly reduced compared with the traditional train plan of full-length and short-turn routes and single train operation scheme. In addition, the train operation plan with virtual coupling trains can improve the maximum and average load factor of trains. Therefore, it proves advantages of the train operation plan of full-length and short-turn routes of virtual coupling trains. Finally, sensitivity analyses are performed using three parameters which include departure frequency of the full-length train and short-turn train, starting and terminal station of the short-turn route, and number of marshalled vehicles of the full-length train and short-turn train. It turns out that all of these parameters have a high impact on the behavior of the value of the objective function.

Selecting the appropriate parameters plays an important role in improving the service level of urban rail transit and reducing operating costs.

When the virtual coupling train is used, the operation staff of the subway can determine a reasonable train operation plan according to the passenger flow of the line, so as to improve the service level and reduce the operating cost of the enterprise. The model proposed in this paper can provide decision support for the operation staff of the subway. The research results of this paper can provide a reference for the optimization research of the train operation plan of full-length and short-turn routes with virtual coupling trains. There are still some aspects to be extended in future work. First, this paper assumes that all stations have the ability to turn back, but this is not the case, so it is necessary to increase the consideration of line conditions. Second, the passengers' choice behavior is a probabilistic problem, and it is necessary to conduct an in-depth analysis of the passengers' choice of train in the mode of full-length and short-turn routes. Third, the full-length train and the short-turn train are coupled and unmarshalled at the turn-back station of the short-turn routing, so how to realize the turnover at the train? These issues need further research in the future.

**Author Contributions:** Conceptualization, X.Z. and F.L.; methodology, X.Z. and F.L.; validation, X.Z.; formal analysis, X.Z. and L.W.; resources, F.L.; data curation, X.Z.; writing—original draft preparation, X.Z., F.L. and L.W.; writing—review and editing, X.Z.; funding acquisition, F.L. All authors have read and agreed to the published version of the manuscript.

**Funding:** This research was supported by Beijing Natural Science Foundation (L201013).

**Institutional Review Board Statement:** Not applicable.

**Informed Consent Statement:** Not applicable.

**Data Availability Statement:** The data presented in this study are available on request from the corresponding author.

**Acknowledgments:** The authors thank everyone who contributed to this article.

**Conflicts of Interest:** The authors declare no conflict of interest.

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
