# Peer review of "Optimization of Train Operation Planning with Full-Length and Short-Turn Routes of Virtual Coupling Trains"

_applsci, doi:10.3390/app12157935_

Round 1
Reviewer 1 Report
Title : Optimization for Train Operation Plan with Full-length and Short-turn Routes of Virtual Coupling Trains in Urban Rail Transit
Authors : Xu Zhou, Fang Lu and Liyu Wang
-----------------------
In this paper, the authors present the train operation plan optimization with virtual coupling trains. A two-level optimization model is proposed with full-length and short-turn routes.
Strengths:
. The abstract presents correctly and synthetically the paper.
. Section 3 is very interesting and detailed with the equations and their meaning. Figure 2 is very illustrative and interesting.
. Conclusion section is clear and synthetic, efficient.
Weaknesses:
. The problem presentation (in section 2) is too dry. You could more explain what problem you are actually solving in the paper.
. Section 4 (solution algorithms) is rather weak. You claim (steps 1 to 3) that "The upper model can be solved using multi population genetic algorithm". Ok, but how? What are the parameters of your genetic algorithm? How do you design the chromosomes? Then, how do you manage the crossing and mutation operations?
. Only one example is proposed in section 5, so that the readers may believe that this paper is only a study done on (undefined) "metro line M", and may not be used anywhere else. I believe the contrary, but invite the authors to prove it with incremental application examples.
Originality / Novelty / Significance:
. The question set in this paper is not original, but well defined.
. The scientific content of this paper could be improved, but the subject addressed is interesting.
. The hypotheses are correctly identified as such. The results could be better presented. This would emphasize the quality of the presented work.
. The technical quality of this paper could be improved, particularly with improving the description of the optimization algorithms.
. The limits of the results obtained in this paper are not mentioned. This point could be investigated. Anyway, I took interest and pleasure to read this paper.
. The conclusion is clear and coherent, but insufficiently justified and supported by the results.
Quality of presentation:
. The abstract is clear and presents correctly the subject addressed in this paper.
. This paper contains the basic sections of a scientific paper. As well, the subheadings used for the redaction of this paper make it clearer.
. This paper is clear, easy to follow and to read, and logically written.
. The data are appropriately presented. Analyses should be more consistent.
. The conclusion is clear enough.
Scientific soundness:
. The subject addressed in this paper is relevant.
. The model has been correctly designed, and is technically sound.
. The results lack of analysis.
. The data presented in this study seems weak to warrant enough robustness to the conclusion.
Overall evaluation:
. The English language quality and style of this paper are appropriate and understandable.
. I think there is an overall benefit to publish this work, after having answered to the suggestions I propose in this review.
. This work provides an advance towards the current knowledge, especially with the model presented, clearly highlighted in the abstract.
. In my opinion, the model, the optimization method and the conclusions of this paper seem to be interesting for the readership of the journal.
. The authors have addressed a relatively easy question, with too weak experiments.
As a conclusion, my suggestion to the editor is to propose to the authors to perform major revisions on their paper before publication in Applied Sciences.
References :
--------------
. 26 research references, relecvantly chosen, out of which only 1 self-reference, giving a good self-reference ratio of 4%.
. The references are cited in the text adequately and appropriately. Nevertheless, some of them could benefit from better analysis in the literature review.
. The bibliography of this paper is mainly composed of recent references: 3 of them are more than 10 years old, and 23 of them are less than 10 years old.
. Only 4 references out of 26 are not Asian ones. Research community is international, and you could cite more Extra-Asian references.
Typos / Comments / Remarks:
------------------------------------
. Line 140: trainError! Reference source not found., this --> You can quickly fix this.
. What is metro line M ?
. Table 5: The Value of Parameter --> The Value of Parameters.
Author Response
Dear reviewer:
Thanks very much for taking your time to review this manuscript. Thanks also for your affirmation of this paper. I really appreciate all your comments and suggestions! I have provided a point-by-point response to the comments. Please see the attachment.
Xu Zhou
2022.6.26

Reviewer 2 Report
-Generally, the authors should revise the English writing carefully. I suggest to revise the title as follows: “Optimization of Train Operation Planning with Full-length and Short-turn Routes of Virtual Coupling Trains”
-What do you mean by a two-level model? Do you mean bi-level programming model or two-stage stochastic model? The term of two-level model is unusual.
-The first paragraph should talk about the advantages of a bi-level programming model for the train operation planning. What is the sequence of decisions? Who is the leader (who makes the first decision)? Who is the follower (who reacts to the decision leader)? How a bi-level approach is better than a general mixed integer programming model?
-As I understood, you should review the literature of bi-level and multi-level programming for the transportation, supply chains and logistics. I highly ask you reading high-quality papers from well-known authors like Prof. Fathollahi-Fard, Prof. Deniz Aksen, Dr. Mahmoodjanloo and so on.
-The research gaps are unclear and your novelty is vague. You should compare your contributions with existing published works in the literature review.
-Analysis Problem is wrong. Problem analysis is correct or analysis of the proposed problem.
-The description of the problem should be revised to provide a connection of parameters and decision variables in each level. I suggest to present a schematic for your bi-level approach? How the upper-level calls the lower level model? Which parameter of upper-level model is a decision variable in the lower-level model?
-There is no definition of a solution in the GA! What is the solution representation? What is the meaning of feasibility for your solution? How did you consider the impacts of constraints on the optimality and feasibility of a solution? What is the search space? What are the neighborhood procedures, i.e., mutation and crossover operators in the GA? I mean how you do some changes on a solution to generate a new solution?
-I suggest reading bi-level metaheuristics from Prof. Talbi to show that how a connection between the upper-level model and the lower-level model, is done?
-There is no comparison of GA with the exact solver to calculate the optimality gap.
-There is no comparison of GA with other similar metaheuristics to show its efficiency.
-There is no statistical analysis on the GA.
-There is no tuning section of GA.
-No managerial insight is generated from the results.
-The conclusion should be updated by findings, limitations and future research recommendations.
Author Response

(The authors gave the same response as above.)

Reviewer 3 Report
This paper talks about proposing a two-level optimization model to focus on the virtual coupling trains. This study illustrates a case study from somewhere and shows the results using the proposed method. The paper demonstrates the proposed method’s structure, the case study scenarios, and the performance evaluations. This paper is practical. However, I have several questions for the authors to address.
1) Page 2. I have questions about the main contributions of this paper. Any transferability of this project? Does this research fill in some research gaps from existing studies? Please highlight them.
2) Section 3.2.2. Why are there some Chinese characteristics?
3) Page 11. Where did the datasets used in the case study come from? More details should be revealed.
4) Any details about the implication of those results rather than comparing results? In other words, any in-depth discussions rather than describing the results?
5) Any echoes from the existing literature that editors can illustrate on top of the results shown in this study? I still think there is much to be leveraged from existing studies.
6) Again, how could other practitioners take advantage of the findings in this study? Please justify it.
Author Response

(The authors gave the same response as above.)

Round 2
Reviewer 1 Report
Title : Optimization of Train Operation Planning with Full-length and Short-turn Routes of Virtual Coupling Trains
Authors : Xu Zhou, Fang Lu and Liyu Wang
-----------------------
Strengths:
. The authors have made efforts to take into account my remarks/comments, and I thank them for this.
. The corrections have globally improved the paper quality.
. Particularly, the section 2 has gained in consistency and clarity.
Still remaining weaknesses:
. However, no significant change has occured since the first version on section 4.
. Section 4 (solution algorithms) is still rather weak, in spite of your efforts to clarify the description. How many populations do you manage? How many individuates in the populations? What are the values of adjustment coefficients? … This section is really determinant and its redaction needs to be taken care of.
. Conclusion has disappeared.
. As well, Chinese characters still remain in version 2, in spite of your replacement promise.
As a conclusion, my suggestion to the editor is to make major corrections to your paper before possible publication in Applied Sciences.
Author Response
Dear reviewer:
Thanks very much for taking your time to review revised version. You are very careful and found many problems in this article. I apologize for my carelessness. I really appreciate all your comments and suggestions! Please read my reply in the attachment.

Reviewer 2 Report
Thank you for providing the revision. The authors have tried to address my comments. However, the authors must revise it carefully again based on my comments:
-The English writing is still poor. I cannot understand many parts of this paper.
-The concept of two-layer model is still unclear. What is its difference with a standard bi-level programming model? Does upper-level model make the decision after the lower-level model? Do you consider a static Stackelberg decision-making? Who is the decision-maker of upper-level model? Who is the decision-maker of lower-level model?
-I ask the authors to separate the introduction and the literature review. In addition, the authors did not review the following relevant papers:
Bi-level programming for home health care supply chain considering outsourcing. Journal of Industrial Information Integration,
Sustainable and Robust Home Healthcare Logistics: A Response to the COVID-19 Pandemic. Symmetry,
-The research gaps are unclear.
-The solution representation is still unclear.
-There is no validation and comparison of GA.
Author Response
Dear reviewer:
Thanks very much for taking your time to review revised version. I really appreciate all your comments and suggestions! Please read my reply in the attachment.

Reviewer 3 Report
The authors have addressed all of my comments, thank you.
Author Response
Dear reviewer:
Thanks very much for taking your time to review revised version. You are very careful and ask many questions about the article. I really appreciate all your comments and suggestions!